# ATTRIBUTE ALIGNMENT AND ENHANCEMENT FOR GENERALIZED ZERO-SHOT LEARNING

## ABSTRACT

Generalized zero-shot learning (GZSL) aims to recognize both seen and unseen classes, which challenges the generalization ability of a model. In this paper, we propose a novel approach to fully utilize attributes information, referred to as attribute alignment and enhancement (A3E) network. It contains two modules. First, attribute localization (AL) module utilizes the supervision of class attribute vectors to guide visual localization for attributes through the implicit localization capability within the feature extractor, and the visual features corresponding to the attributes (attribute-visual features) are obtained. Second, enhanced attribute scoring (EAS) module employs the supervision of the attribute word vectors (attribute semantics) to project input attribute visual features to attribute semantic space using Graph Attention Network (GAT). Based on the constructed attribute relation graph (ARG), EAS module generates enhanced representation of attributes. Experiments on standard datasets demonstrate that the enhanced attribute representation greatly improves the classification performance, which helps A3E to achieve state-of-the-art performances in both ZSL and GZSL tasks.

## 1 INTRODUCTION

Zero-shot learning aims to recognize unseen classes that have not been appeared during training phase, a common solution resort to auxiliary information to bridge the gap between seen and unseen domains to achieve knowledge transfer from the seen to the unseen. Semantics are the most frequently used auxiliary information for ZSL, either by class descriptions, word vectors (Mikolov et al., 2013) or attributes (Farhadi et al., 2009). A general paradigm (Xie et al., 2019; Zhu et al., 2019; Huynh & Elhamifar, 2020b; Min et al., 2020; Xie et al., 2020; Ge et al., 2021; Liu et al., 2021b; Chen et al., 2021b; 2022) is to learn a mapping that projects visual features of seen samples into an embed-ding space to align with semantic attributes. With the assumption that seen and unseen domains share the same attribute space, the learned knowledge from seen classes is easily transferred to the unseen ones. And then, the subsequent classi-fication is accomplished by measuring compatibil-ity scores between the projected features and the attribute prototypes. Recent works on embeddings turn to local features of image parts, i.e. part-based embedding meth-ods (Elhoseiny et al., 2017), to learn discriminative features easy for classification. Comparatively, gener-ative methods (Xian et al., 2019b; Huynh & Elhamifar, 2020a; Ma & Hu, 2020; Han et al., 2021; Chen et al., 2021a;c; Chou et al., 2021) utilize semantic information of unseen classes to synthesize unseen visual features by a generative model, such as generative adversarial network (GAN) (Goodfellow et al., 2020) or vari-ational autoencoder (VAE) (Kingma & Welling, 2013), so that convert zero-shot classification to the traditional supervised model learning that could be trainable with generated samples. However, the features inferred from semantic information mostly are high-level visual representation, which are often non-discriminative to class recognition (Huynh & Elhamifar, 2020b; Xian et al., 2019b; Huynh & Elhamifar, 2020a).

Recently, generalized zero-shot learning (GZSL) for its rigorous and realistic nature has received increasing attention in this field, where seen classes and unseen classes constitute the testing space. Embedding methods are inherently inferior in GZSL since the model training merely relies on sam-ples of seen classes, and thus inevitably biases towards the seen ones. Moreover, the visual-semantic alignment in embedding models is just operated in seen domain, and the visual di-vergence between seen and unseen domains may strengthen the bias, namely domain shift (Fu et al., 2014). Differ-ent methods have been explored to improve the model performance in GZSL. Some studies try to

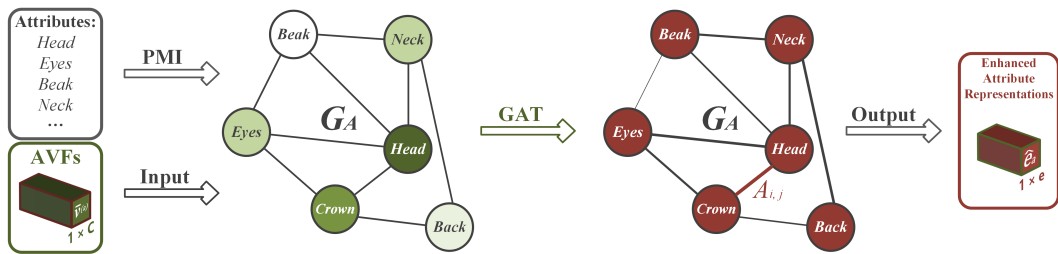

Figure 1: Attribute Localization.

mitigate the bias by introducing constraints on losses to calibrate output pre-diction probability, which usually require unseen semantics as side infor-mation (Huynh & Elhamifar, 2020b; Xie et al., 2020). The Parts Relation Rea-soning is used in RGEN (Xie et al., 2020) to capture appearance relationships among image parts, which is believed to be a complementary cue for improving the performance. GCNZ (Veličković et al., 2018) utilizes class relationships to infer classifier parame-ters directly from knowledge graph. Relation learning is no novelty to ZSL, however, the semantic relationship between attributes is rarely explored in previous works. Huynh & Elhamifar (2020b) have informed us by introduc-ing word vectors of attribute that there is a wealth of semantic infor-mation in attributes beyond the commonly used class attribute vectors. There are also rich semantic relationships between attributes, which can be transferred to visual domain to help mitigate visual-semantic gap. Once the relations between attributes are modeled, it is possible to enhance the fi-nal prediction of classes by the interplay of attributes. Existing methods tried to capture the semantic relations in the at-tributes, such as using the entanglement of CNN and GCN based on knowledge graph about attrib-utes (Hu et al., 2022). However, despite the fact that nodes in graph are explic-itly defined as attributes, those methods lack a mechanism to accurately align nodes to the corresponding attributes. To the best of our knowledge, the fusion of relation learning and attention mechanism has not been studied in ZSL.

Accordingly, we propose an attribute alignment and enhancement (A3E) network for GZSL, which incorporates attribute alignment (AA) pipeline and attribute enhancement (AE) module. AA pipeline consists of attribute localization (AL) module and attribute scoring (AS) module, this novel ap-proach of attribute alignment allows model to subtly catch visual features corresponding to attributes (namely attribute-visual features, AVFs) by fully utilization of attribute knowledge (both class at-tribute vectors and attribute word vectors). Compared to previous part-based methods that require complex accessories such as attention module and part detector, A3E simplifies its AA pipeline to a single convolutional layer with a single linear transformation, and still delivers competitive re-sults. Most importantly, the resulted AVFs serve as the carriers for attributes which support the subsequent attribute enhancement process. In order to model the relations of attributes, AE module first constructs an attribute-relation graph (ARG), where relation-ships of attributes are quantified as graph edges, then, facilitated by graph neural networks, embeds the input AVFs into attribute semantics space. The enhanced attribute features are obtained through the outputs of graph nodes. Figure 1 demonstrates the basic process of AE module. Experiments in three standard ZSL datasets show that A3E reaches the state-of-the-art results in both ZSL and GZSL without extra information from unseen classes or auxiliary constraints on output probabilities, verifying the advantages of our proposed method.

Our contributions can be summarized as:

• A novel attribute enhancement (AE) module is created to explicitly model the relationship between attributes, and the enhanced attribute representation is generated with attribute-relations modeled inside.

• To align graph nodes with attributes, an efficient attribute alignment (AA) pipeline is designed to generate visual fea-tures corresponding to attributes, namely attribute-visual features (AVFs).

• We propose an attribute alignment and enhancement (A3E) network that based on the AA pipeline and AE module, an innovative combination of attention mechanism and semantic-relation learning.

Extensive experiments on three bench-marks show that our design can significantly improve results in both ZSL and GZSL tasks.

## 2 RELATED WORK

There are two main paradigms for ZSL/GZSL: generative methods (Xian et al., 2019b; Huynh & Elhamifar, 2020a; Ma & Hu, 2020; Han et al., 2021; Chen et al., 2021a;c; Chou et al., 2021) and embedding methods (Xie et al., 2019; Zhu et al., 2019; Huynh & Elhamifar, 2020b; Min et al., 2020; Xie et al., 2020; Ge et al., 2021; Liu et al., 2021b; Chen et al., 2021b; 2022). Generative methods covert ZSL problem into traditional supervised learning using visual features synthesized by generative models for unseen classes (Liu et al., 2021a). However, generative models such as GAN or VAE are often difficult to generate high-quality synthetic samples for unseen classes to train classifiers (Pourpanah et al., 2022). On the other hand, embedding methods learn a mapping that aligns visual features with semantic prototypes, therefore achieve knowledge transfer from seen to unseen classes via their sharable semantics. According to the mapping space, embedding methods can be divided into three categories: visual space embedding (Zhang et al., 2017), semantic space embedding (Zhu et al., 2019; Huynh & Elhamifar, 2020b; Xie et al., 2020; Liu et al., 2021b) and common space embedding (Min et al., 2020), with their respective pros and cons.

As its name suggests, semantic space embedding projects visual features into semantic space. Recent studies further suggest that global visual features are detrimental to classification (Xie et al., 2019; Zhu et al., 2019; Huynh & Elhamifar, 2020b; Xie et al., 2020). Instead of using noisy global features, part-based embedding methods try to improve classification performance by locating discriminative parts in image. Elhoseiny et al. (2017) deployed a visual part detector to link text descriptions with corresponding image regions, which would be fed into the part-based visual classifiers. SGMA (Zhu et al., 2019) employed a multi-attention module and DAZLE (Huynh & Elhamifar, 2020b) constructed a hierarchical linear structure, all in order to focus the model on discriminative regions in image. Whereas, the model with part detector attention module would become complex, so that make it difficult to train and optimize. SELAR (Yang et al., 2021) proposed to localize part features by the implicit localization ability within feature extractor, where the complex attention module is replaced with a single convolution layer. Most of the above models use class attribute vectors as semantic information. However, since the attribute space spanned by class attribute vectors is inevitably suffered from hubness problem (Zhang et al., 2017), the choice of embedding space is still an issue that is worth to explore in subsequent study.

With the increasing attention paid to ZSL, various techniques from other fields were also incorporated into ZSL models, such as knowledge distillation (Chen et al., 2022), meta-learning (Verma et al., 2020) and graph learning (Xie et al., 2020; Wang et al., 2018). Graph Neural Networks (GNNs) (Kipf & Welling, 2017; Veličković et al., 2018) were proposed to model non-Euclidean data, especially for those with graph structure. Veličković et al. (2018) firstly introduced Graph Convolutional Networks (GCN) (Kipf & Welling, 2017) to explicitly model relations between classes in ZSL by knowledge graph. And RGEN (Xie et al., 2020) employed GCN to represent the relations among local image regions. Whereas, none of them have explored the semantic relations that implied within attributes.

Inspired by the advantages and deficiencies of previous works, our proposed A3E employs a dual embedding strategy to fully utilize the rich semantics beneath attributes, and incorporates GAT (Veličković et al., 2018) to dynamically model the semantic relations between attributes.

## 3 ATTRIBUTE ALIGNMENT AND ENHANCEMENT

In this section, we will specify how and why the A3E network is proposed. Here we follow the pipeline that A3E processes the samples, and present the whole structure and details of our model.

### 3.1 ATTRIBUTE LOCALIZATION

In order to explore those rich semantics and relations between attributes, we need to obtain the visual representations for attributes first. Instead of generating discriminative regions using various of attention modules, Yang et al. (2021) innovated to utilize the implicit attribute localization ability

within feature extractor (CNNs) to get part location, here we refer to it as attribute localization (AL). AL greatly reduces the complexity of the model by replacing the complicated attention module with a single 1×1 convolution:

$$\tilde{\mathbf{a}} = conv\left(\mathbf{v}\right) \tag{1}$$

where, $\mathbf{v} = \varphi\left(\mathbf{x}_i\right)$ is the visual features with $H \times W \times C$ dimensions extracted by the backbone network $\varphi\left(\cdot\right)$. $\mathbf{x}_i$ is the $i-$th input image, and $conv\left(\cdot\right)$ is the 1×1 convolution with $1 \times 1 \times C \times A$ parameters. $\tilde{\mathbf{a}} \in \mathbb{R}^{H \times W \times A}$ is the output features with attribute localization, referred to as attribute features.

Under the supervision of attribute vectors, this simple convolution could gather most important spacial information of attributes. The loss function of AL module is defined as follows:

$$\mathcal{L}_{AL} = \mathcal{CE}\left(SoftMax\left(\mathbf{A}^S GMP(conv\left(\mathbf{v}\right))^T\right), y_i\right) \tag{2}$$

where $y_i$ is the label of $i-$th input image, and $GMP\left(\cdot\right)$ is the global maximum pooling function that employs spatial aggregation to the attribute features. $\mathbf{A}^S \in \mathbb{R}^{N^S \times A}$ is the seen attributes matrix where $N^S$ is the number of seen classes. $SoftMax\left(\cdot\right)$ is SoftMax activation function and $\mathcal{CE}\left(\cdot\right)$ is the cross entropy loss commonly used in ZSL models.

Blue area at the bottom of Figure 2 shows the layout of AL module.

## 3.2 ATTRIBUTE SCORING

As mentioned in the previous section, attribute space, despite being commonly used in ZSL models, suffers from several problems like hubness problem. We are aware of the rich semantic information beneath attributes. Inspired by Huynh & Elhamifar (2020b), we introduce attribute semantic space to collaborate with attribute space, which forms our attribute scoring (AS) module.

With the attribute features from AL module, the visual location of every attribute is encoded in each channel of $\tilde{\mathbf{a}}$. Naturally, we thought of making it into a set of attention masks as a substitute for attention mechanism. The attribute masks are obtained through the sigmoid function that normalizes values of each channel in $\tilde{\mathbf{a}}$ into a range from 0 to 1, where the value approaching to 1 stands for high confidence of having attribute-related visual features in the location, while that approaching to 0 is the opposite. Therefore, we can extract visual features for each attribute-related image region using attribute masks by performing the broadcasted Hadamard production between visual features $\mathbf{v} \in \mathbb{R}^{H \times W \times C}$ and each channel of normalized $\tilde{\mathbf{a}}$, which produces A masked visual features that correspond to A attributes, namely attribute-visual features (AVFs) $\mathbf{v}^{(a)} \in \mathbb{R}^{H \times W \times C}$ $(a \in [1, A])$.

To achieve zero-shot classification, the next step is to map AVFs into attribute semantic space with reference to DAZLE (Huynh & Elhamifar, 2020b). The attribute semantic space is constructed using word vectors of attributes that are usually produced by word vector models like Word2Vec (Mikolov et al., 2013). With these attribute word vectors, we employ AS module to map the above AVFs into attribute sematic space, and then calculate the class score of samples as follows:

$$\hat{\mathbf{e}}_a = \mathbf{W}\left(GAP\left(\mathbf{v}^{(a)}\right)\right), a \in [1, A] \tag{3}$$

$$p_a = \mathbf{e}_a \hat{\mathbf{e}}_a^T, a \in [1, A] \tag{4}$$

$$s^c = \mathbf{a}^c \mathbf{p}^T, c \in \left[1, N^S\right] \tag{5}$$

where $GAP\left(\cdot\right)$ is the global average pooling function that aggregates AVFs into $1 \times C$ dimensions, and $\mathbf{W}\left(\cdot\right)$ is the linear mapping with $1 \times C \times e$ parameters. $\hat{\mathbf{e}} \in \mathbb{R}^{1 \times e}$ is the predicted word vector, and $\mathbf{e} \in \mathbb{R}^{1 \times e}$ is the ground truth attribute word vector. $p_a$ is the attribute score for the $a-$th attribute. $\mathbf{p} \in \mathbb{R}^{1 \times A}$ is the concatenation of $p_a$ $(a \in [1, A])$. $\mathbf{a}^c \in \mathbb{R}^{1 \times A}$ $\left(c \in \left[1, N^S\right]\right)$ is the class attribute vector of the $c-$th class which is extracted from the $c-$th row of attribute matrix $\mathbf{A}^S$. And $s^c$ is the class score of the current sample belonging to the $c-$th class.

Thus, the attribute scoring loss based on cross entropy is designed:

$$\mathcal{L}_{AS} = \mathcal{CE}\left(SoftMax\left(\mathbf{s}_{AS}\right), y_i\right) \tag{6}$$

where $\mathbf{s}_{AS} \in \mathbb{R}^{N^S}$ is the concatenation of class score $s^c \left(c \in \left[1, N^S\right]\right)$.

Ultimately, a novel attribute alignment (AA) pipeline is constructed, which consists of AL and AS modules. AA innovates attribute alignment approach through integrating attribute space and attribute semantic space into a unified pipeline of ZSL model. Subsequently, we will model the relations between attributes.

### 3.3 ATTRIBUTE ENHANCEMENT

Studies on CNNs (or DNNs) have shown that feature extractor built up by convolutional neurons does well in extracting visual patterns from images. However, what they are not good at is to extract non-visual concepts from image samples. Unfortunately, the abstract concepts are common in the attribute sets of many ZSL datasets. For example, Animals with Attributes 2 (AwA2) (Xian et al., 2019a) has 85 expert-defined attributes in total. Roughly half of these attributes can be directly related to visual representations (like "stripes" and "tail"), while more than half of them do not correspond directly to the visual representation (such as "fast" and "smart"). This is even more troublesome for part-based ZSL methods since it is hard to locate a visual region for such non-visual attributes thus confusing the model.

To address the issue, we turn to exploiting the implicit semantic relations between attributes. The idea is to dynamically build the associations between visual-related attributes and non-visual attributes by modeling the relations between them. With the assumption that all attributes share the same semantic space, it allows the model to enhance the usability of non-visual attributes using representations of visual-related attributes.

In order to model the relations of attributs, we construct an attribute relation graph (ARG) $\mathbf{G}_A$ based on point-wise mutual information (PMI) (Bouma, 2009) according to Hu et al. (2022). Let $\mathbf{G}_A$ has A vertices corresponding to A attributes, the edges between attributes (vertices) are defined based on their normalized PMI values referring to threshold $\delta$:

$$A_{i,j} = A_{j,i} = \left\{ \begin{array}{l} 1, PMI_n\left(x, y\right) > \delta \\ \quad 0, else \end{array} \right. \tag{7}$$

where $A_{i,j}$ is the element in the $i-$th row and $j-$th column of $\mathbf{A}$. We use the undirected graph for ARG, so that its adjacency matrix is symmetric, that is $A_{i,j} = A_{j,i}$. See Appendix A for the detailed formula of PMI. The selection of threshold $\delta$ will be discussed in detail in experiments section.

So far, ARG discussed above is still a static structure shared by all categories, which cannot be dynamically optimized with different classes and samples. Meanwhile, the PMI-based connections (graph edges) may not always represent the correct relations between attributes. Therefore, instead of using the well-known graph convolutional networks (GCN), we leverage graph attention networks (GAT) (Veličković et al., 2018) to achieve dynamic modeling based on ARG.

GAT proposed by Veličković et al. can dynamically adjust edge weights with self-attention mechanism to solve a series of problems in spatial GNNs. According to AA pipeline in the previous subsection, A sets of global AVFs $\bar{\mathbf{v}}^{(a)} \left(a \in [1, A]\right)$ belonging to the sample $x_i$ are obtained, which will be subsequently used as the input of nodes in $\mathbf{G}_A$. We use a two-layer GAT network in the paper, i.e. $l \in \{0, 1\}$. The outputs of GAT are the predicted word vectors $\hat{\mathbf{e}}_a \in \mathbb{R}^{1 \times e} \left(a \in [1, A]\right)$ from each node, where $e$ is the dimension of word vector. In the process, GAT not only projects the input global AVFs into attribute semantic space (like the AS module), but also models attribute relations in the output, namely predicted word vectors $\hat{\mathbf{e}}_a$. With the help of ARG that connected attributes by their semantic relation, GAT can enhance the expression of certain attribute-related features (especially those from non-visual attributes), yielding more discriminative semantic representations. Hence, we name it attribute enhancement (AE) module.

As the output of AE module has the same form of predicted word vectors like AS module, an integrated enhanced attribute scoring (EAS) module is naturally formed. We can follow formula

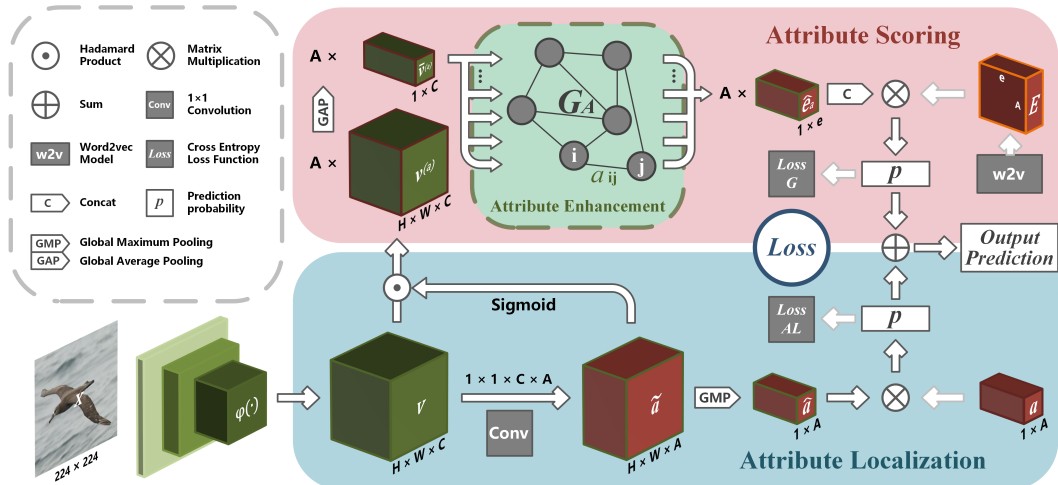

Figure 2: Attribute Alignment and Enhancement Network.

(3), (4) and (5) to calculate the class score vector $\mathbf{s}_{EAS}$, and the enhanced attribute scoring loss is constructed as:

$$\mathcal{L}_{EAS} = \mathcal{CE}\left(SoftMax\left(\mathbf{s}_{EAS}\right), y_i\right) \tag{8}$$

The dropout strategy is often used to improve the generalization of GNN models by randomly "dropping" some nodes (sweeping node features) during training. The method is originally used in GNNs to suppress over-smoothing problem (Li et al., 2018), while we use here to perturb the original data (AVFs) to enhance the generalization of the model. Subsequent experiments will demonstrate the significant impact of the dropout strategy on GZSL performance.

### 3.4 GENERALIZED ZERO-SHOT IMAGE CLASSIFICATION

Finally, by integrating the attribute localization (AL) module and enhanced attribute scoring (EAS) module, A3E network for GZSL is constructed (see Figure 2).

#### 3.4.1 OVERALL LOSS

The overall loss function of A3E is:

$$\mathcal{L}_{A3E} = \lambda\mathcal{L}_{AL} + (1 - \lambda)\mathcal{L}_{EAS} \tag{9}$$

where $\lambda$ is the weighting coefficient that balance the two modules.

#### 3.4.2 INFERENCE

In the inference stage, A3E employs a fusion prediction method as follows:

$$\hat{y} = \arg\max_{c \in N}\left(\lambda\mathbf{s}_{AL} + (1 - \lambda)\mathbf{s}_{EAS} + \beta\Delta_{[c \in N^U]}\right) \tag{10}$$

where $\hat{y}$ is the predicted label, and $\lambda$ is a coefficient which has identical function with the one in training loss. $\mathbf{s}_{AL}$ is the output probability of AL module and $\mathbf{s}_{EAS}$ is the output probability of EAS module. $N = N^S \cap N^U$ is the set of all classes labels. $\beta$ is an adjustable bias, and $\Delta_{[c \in N^U]}$ is an indicator which will take 1 for unseen classes, and -1 for seen classes.

To alleviate the inevitable bias caused by domain shift in GZSL, we set a calibration bias refer to (Huynh & Elhamifar, 2020b). However, it is worth notice that unlike (Huynh & Elhamifar,

Table 1: Results of conventional ZSL and GZSL classification on AwA2, CUB and SUN datasets. The best and second-best results are marked in bold and underline, respectively. The symbol "-" indicates no results. The symbol "*" represents models with 448×448 input size.

| Methods | | AwA2 | | | | CUB | | | | SUN | | | |
|---|---|---|---|---|---|---|---|---|---|---|---|---|---|
| | | ZSL | GZSL | | | ZSL | GZSL | | | ZSL | GZSL | | |
| | | MCA | Unseen | Seen | H | MCA | Unseen | Seen | H | MCA | Unseen | Seen | H |
| **Generative Methods** | | | | | | | | | | | | | |
| CVPR 2019 | f-VAEGAN-D2 | 71.1 | 57.6 | 70.6 | 63.5 | 61.0 | 48.4 | 60.1 | 53.6 | _64.7_ | 45.1 | _38.0_ | 41.3 |
| NeurIPS 2020 | Composer | 71.5 | 62.1 | 77.3 | 68.8 | 69.4 | 56.4 | 63.8 | 59.9 | 62.6 | **55.1** | 22.0 | 31.4 |
| ICCV 2021 | FREE | - | 60.4 | 75.4 | 67.1 | - | 55.7 | 59.9 | 57.7 | - | 47.4 | 37.2 | _41.7_ |
| **Embedding Methods** | | | | | | | | | | | | | |
| NeurIPS 2019 | SGMA | 68.8* | 37.6* | **87.1*** | 52.5* | 71* | 36.7* | 71.3* | 48.5* | - | - | - | - |
| CVPR 2020 | DAZLE | - | 60.3 | 75.7 | 67.1 | 65.9 | 56.7 | 59.6 | 58.1 | - | _52.3_ | 24.3 | 33.2 |
| ECCV 2020 | RGEN | _73.6_ | _67.1_ | 76.5 | **71.5** | 76.1 | 60.0 | 73.5 | 66.1 | 63.8 | 44.0 | 31.7 | 36.8 |
| AAAI 2021 | SR2E | - | 58* | _80.7*_ | 67.5* | - | 61.6* | 70.6* | 65.8* | - | 43.1* | 36.8* | 39.7* |
| SPL 2021 | SELAR | - | 52.0 | 71.9 | 60.3 | - | 62.4 | 64.9 | 63.6 | - | 40.5 | 32.9 | 36.3 |
| NeurIPS 2021 | HSVA | - | 56.7 | 79.8 | 66.3 | 62.8 | 52.7 | 58.3 | 55.3 | 63.8 | 48.6 | **39.0** | **43.3** |
| CVPR 2022 | MSDN | 70.1* | 62* | 74.5* | 67.7* | **76.1*** | **68.7*** | 67.5* | _68.1*_ | **65.8*** | 52.2* | 34.2* | 41.3* |
| ours | A3E | **74.1** | **69.3** | 71.2 | _70.2_ | _74.9*_ | _66.3*_ | _71.6*_ | **68.8*** | 64.0* | 46.8* | 30.0* | 36.5* |

2020b), the calibration in our method only includes a bias on the prediction probability, and does not include any additional loss term that usually requires unseen semantics to adjust output predictions.

# 4 EXPERIMENTS

## 4.1 EXPERIMENTAL SETUP

### 4.1.1 DATASETS

We evaluate A3E network on three GZSL benchmarks: Animals with Attributes 2 (AwA2) (Xian et al., 2019a), Caltech-UCSD Birds 200-2011 (CUB) (Wah et al., 2011) and SUN attribute database (SUN) (Patterson et al., 2014). AwA2 is a coarse-grained dataset with 37,322 images from 50 animal classes, each of which has 85 attributes. While, CUB is a fine-grained bird dataset containing 11,788 images from 200 bird classes with 312 attributes. SUN is also a fine-grained dataset which includes 14,340 images from 717 scene categories, and each class has a 102-dimension attribute vector.

In order to fairly compare with the state-of-the-art ZSL models, we adopt the proposed split (PS) of datasets presented in (Xian, Schiele, and Akata 2017). Evaluation metrics are shown in Appendix B

### 4.1.2 IMPLEMENTATION DETAILS

All the experiments in the paper are conducted on NVIDIA GeForce RTX 3090 with 24 GB video memory size. Software versions are Python 3.9, PyTorch 1.11.0, NumPy 1.22.3 and CUDA 11.3.

For better reproducibility of experiment results, we manually fixed the random seed to 1024 for all tests. Detailed settings would be listed in Appendix C

## 4.2 COMPARISON WITH STATE-OF-THE-ARTS

We compare our A3E network with several state-of-the-art models in both ZSL setting and GZSL setting. The results are presented in Table 1. On the ZSL side, A3E shows very gratifying results in all three datasets. While it is competitive compared with the state-of-the-art models on CUB and SUN, the most compelling achievement is that A3E outperforms the long-standing record by RGEN (Xie et al., 2020) on AwA2 since 2020. On the GZSL side, A3E reaches the highest harmonic mean (68.8%) on CUB dataset, which is the current best generalized model on CUB. It also gets a satisfactory performance on AwA2, though is inferior to RGEN. Whereas, harmonic mean of A3E on AwA2 is still improved by at least 2.5% compared with the former state-of-the-art models, which verifies that A3E is a strong competitor so far, compared with most of the models except for RGEN. In general, the results prove that the proposed attribute alignment and enhancement work effectively on AwA2 and CUB, which are beneficial to represent the rich semantic relations between attributes

Table 2: Ablation study under two datasets. The best results are marked in bold.

| Method | AwA2 | | | | CUB | | | |
|---|---|---|---|---|---|---|---|---|
| | ZSL | GZSL | | | ZSL | GZSL | | |
| | MCA | Unseen | Seen | H | MCA | Unseen | Seen | H |
| AL+AS(AA pipeline) | 60.4 | 52.6 | **78.5** | 63.0 | 57.7 | 44.6 | 60.5 | 51.3 |
| AL+EAS(A3E Network) | **74.1** | **69.3** | 71.2 | **70.2** | **74.9** | **66.3** | **71.6** | **68.8** |

hidden in AwA2 and CUB. The data characteristic of AwA2 and CUB promotes A3E network more generalized. However, the performance on SUN dataset declines sharply. The reason accounted for the phenomenon is that attributes of SUN are more abstract which are hard to capture their correspondent image regions. More importantly, the semantic relations are too sparser compared with the former datasets to affect the performance of attribute enhancement.

Among all comparison models, DAZLE, RGEN and MSDN are the most noteworthy as well as our A3E network. They have some similarity in research motivation. DALZE and MSDN adopt attribute scoring mechanisms, as well as A3E. RGEN uses GNN to perform region-based relations on visual features. Whereas, A3E employs AE module to model semantic relations between attributes. From the GZSL results on AwA2 and CUB, A3E successfully surpasses these models by the stable performance which can be expressed as the average of harmonic means: A3E network reaches 69.5% in average, while DAZLE, RGEN and MSDN are 62.6%, 68.8% and 67.9%, respectively(see Appendix D). The more surprising fact is that when A3E generalizes well in GZSL settings, it does not resort to any probability tricks commonly used in above models, such as the balance loss used by RGEN and the calibration loss used in DALZE and MSDN models. The modified losses by the probability tricks require the supervision from unseen semantics during training, which are contrary to the original setting of ZSL to some extent. In contrast, A3E only relies on seen samples and generalizes better than those models that require unseen semantics, which further demonstrates the superiority of the propose method.

## 4.3 ABLATION STUDY

### 4.3.1 EFFECTS OF ATTRIBUTE ENHANCEMENT

The proposed A3E network is composed of three interrelated modules, namely, AL, AS and AE modules. Among them, AE module based on GAT is the most innovative and representative component of our work. To further evaluate the efficacy of attribute enhancement in actual task, we conduct ablation studies on AwA2 and CUB datasets, by setting the baseline model with only AL and AS modules, i.e. attribute alignment (AA) pipeline. Table 2 shows the results of ablation study.

To fairly compare the baseline model with A3E network, we use the same parameter setting in both. Even so, the ZSL accuracy of A3E has improved by staggering 13.7% and 17.2% for AwA2 and CUB datasets, respectively. Similarly, drastic performance boost is also present in the GZSL setting, where A3E exceeds the baseline by up to 7.2% and 17.5% in harmonic mean metric on AwA2 and CUB, respectively. Thus, with the help of AE module, A3E exceeds the baseline with absolute superiority in all settings, with an even greater advantage in CUB. In effect, the characteristics of attributes such as semantics and relations vary with different datasets. The reason that the performance on CUB has dramatic improvement by AE module is resulted from the stronger semantic relations between attributes of CUB, where attributes are more uniform to describe image contents.

### 4.3.2 EFFECTS OF DROPOUT

To investigate the effects of dropout in AE module, we conduct experiments with different dropout rates on CUB dataset. The results are concluded in Appendix E.

(a) results on AwA2    (b) results on CUB    (a) results on AwA2    (b) results on CUB

Figure 3: Effects of weighting coefficient $\lambda$ on AwA2 and CUB datasets.

Figure 4: Effects of threshold $\delta$ on AwA2 and CUB datasets.

## 4.4 HYPERPARAMETER ANALYSIS

### 4.4.1 EFFECTS OF WEIGHTING COEFFICIENT $\lambda$

Figure 3 is the summary of the experimental results on tuning weighting coefficient $\lambda$ between AL and EAS modules. We do not show experiments for $\lambda$ at 0 and 1 since they are no longer considered in our A3E model. In CUB dataset (see Figure 3(b)), both ZSL and GZSL indicators increase with the rise of $\lambda$. While in AwA2 dataset (see Figure 3(a)), the accuracies of unseen classes in ZSL and GZSL rise with the increase of $\lambda$. But the accuracy of seen classes in GZSL shows the opposite trend, so as to cause harmonic means of GZSL slowly improving. The increasing of $\lambda$ represents that the model emphasizes more on AL module learning. Results show that despite the simpler structure and fewer parameters of the AL module, its importance in the objective function is no less than that of the EAS module. It is proven by the results that model performance improves as the effort invested in AL module (i.e., value of $\lambda$) increases. Specially, attribute localization is more demanding for the fine-grained datasets with more subtle features, such as the CUB dataset. Considering all the indicators, we set $\lambda$ to 0.6 and 0.9 for AwA2 and CUB, respectively.

### 4.4.2 EFFECTS OF THRESHOLD $\delta$

Threshold $\delta$ controls the generation of ARG. Smaller value of $\delta$ indicates easier connection between nodes in ARG, which means more edges in the graph. When $\delta = 1$, there is no edge in ARG, so that AE module does not work. Figure 4 shows how different values of $\delta$ affect model performance on the two datasets. As $\delta$ increases in CUB dataset (see Figure 4(b)), all four metrics increase equally until $\delta = 0.8$. In AwA2 dataset (see Figure 4(a)), we can find a general uptrend in the accuracy of unseen classes as $\delta$ increase, and the best accuracy is found at $\delta = 0.9$. With the increase in $\delta$, the edges representing attribute relationships in ARG should gradually decrease. Obviously, edges created by PMI do not perfectly correspond with the semantic relationships of attributes. We believe that the graphs generated by lower values of $\delta$ have more noisy connections (edges), which leads to the model performance decline. When the structure of ARG is simplified with higher threshold, GAT is more likely to obtain robust information from ARG, thus enhancing the classification performance. To balance the accuracies of seen classes and unseen classes, we set $\delta = 0.9$ and $\delta = 0.8$ for AwA2 and CUB, respectively.

## 5 CONCLUSION

In the paper, we propose an attribute alignment and enhancement (A3E) network, which consist of attribute alignment (AA) pipeline and AE module. Therefore, A3E can align each attribute with corresponding image region and enhance their representations by the semantic relations between attributes through GNNs. At last, the experiments on three ZSL datasets have demonstrated the superiority of A3E network on ZSL/GZSL classification.

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

## A  POINT-WISE MUTUAL INFORMATION

Point-wise mutual information (PMI) (Bouma, 2009) is proposed measure the relationship of two points (objects), defined as:

$$PMI(x, y) = \ln \frac{p(x, y)}{p(x)\,p(y)} \tag{11}$$

where $p(x, y)$ is the joint probability of $x$ and $y$.

In practice, the normalized PMI is more commonly used as:

$$PMI_n(x, y) = \frac{PMI(x, y)}{-\ln p(x, y)} \tag{12}$$

where its values are in the range of [-1,1], which give a good indication of the relevance between $x$ and $y$. Specifically, 1 means they are co-occurrence, while -1 shows the opposite case. But, 0 indicates they are not relevant at all.

## B  EVALUATION METRICS

We consider both conventional ZSL and GZSL settings in all three datasets. Mean Class Accuracy ($MCA$) is adopted as the evaluation indicator for ZSL setting, which takes average value of Top-1 accuracies on unseen classes. And harmonic mean ($H$) is employed to evaluate the performance of GZSL setting, which is the most comprehensive metric to reflect model performance by taking accuracies of seen and unseen classes into consideration. The specific formula is defined as follows:

$$H = \frac{2 \times MCA_S \times MCA_U}{MCA_S + MCA_U} \tag{13}$$

where $MCA_S$ and $MCA_U$ are the $MCAs$ for seen classes and unseen classes, respectively.

## C    MODEL SETTINGS

We use the fixed ResNet101 (He et al., 2016) pretrained on ImageNet as the feature extractor of A3E network, which is commonly used as the backbone network in many models (Huynh & Elhamifar, 2020b; Xie et al., 2020; Ge et al., 2021; Chen et al., 2021b; 2022). The input images of model are reshaped as 224×224 pixels for AwA2 datasets and 448×448 pixels for CUB and SUN datasets since the finer details could significantly improve performance on fine-grained datasets. We use the Word2Vec model trained on Google News to generate attribute word vectors with 300 dimensions.

We adopt ADAM optimizer (Kingma & Ba, 2017) in model training and set weigh decay to $1\times10^{-5}$. We empirically set the hidden layers and attention heads of GAT network in EAS module to $\{200, 1\}$ for AwA2, $\{200, 4\}$ for CUB and $\{1000, 5\}$ for SUN, with dropout rate fixed to 0.2. For AwA2 dataset, we set the learning rate to $5 \times 10^{-6}$, batch size to 64 and maximum iteration number to 10. Regarding to CUB dataset, the learning rate is set to $7.5 \times 10^{-6}$. Batch size is set to 8, and maximum iteration number is 30. As to SUN dataset, we set the learning rate to $5 \times 10^{-6}$, batch size to 16 and maximum iteration number to 25. The learning rate for 1×1 convolution in AL module is 10 times greater than the given values in all datasets. The calibration bias $\beta$ is set to 2.0, 0.4 and 0.4 for AwA2, CUB and SUN, respectively.

There are two hyperparameters in A3E network: weighting coefficient $\lambda$ and ARG threshold $\delta$. We set $\lambda$ to 0.6, 0.8 and 0.9 for AwA2, CUB and SUN datasets. While $\delta$ is set to 0.9, 0.8 and 0.1 for these three datasets, respectively. The influence of the hyperparameters is explored in the following experiments.

## D    AVERAGE HARMONIC MEANS ON AWA2 AND CUB

Table 3: Average harmonic means on AwA2 and CUB datasets.

| Method | AwA2 | CUB | Average |
|---|---|---|---|
| | H | | |
| DAZLE | 67.1 | 58.1 | 62.6 |
| RGEN | **71.5** | 66.1 | 68.8 |
| MSDN | 67.7 | 68.1 | 67.9 |
| A3E | 70.2 | **68.8** | **69.5** |

## E    ABLATION STUDY ON DROPOUT

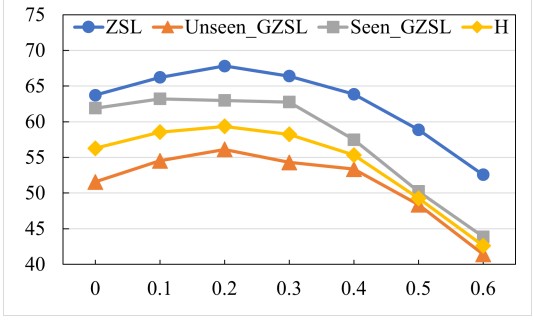

Figure 5: Effects of dropout rate $d$ on CUB dataset.

Dropout is a commonly used strategy to prevent neural networks from overfitting (Srivastava et al., 2014). In our work, we employ dropout on GAT of AE module by randomly dropping some node features during training stage. The dropping process is controlled by dropout rate $d$ that determines the proportion of dropped nodes. As shown in Figure 5, with dropout rate $d$ increasing from 0 to 0.2 (0 means dropout is inactive), almost all indicators are on the rise, except a slight fluctuation

for the seen class accuracy on GZSL. The accuracy decreases gradually from 0.3. To sum up, both ZSL accuracy and GZSL harmonic mean reach the best values at 0.2 dropout rate, then begin to decline with the increase of $d$. The phenomenon can verify functionality of dropout: preventing models from overfitting. Besides, dropout is indispensable to AE module because it improves the generalization ability of model.

## F  TERM EXPLANATION

Attribute semantics: The word vectors of attributes, which is the output of word2vec model by taking the attribute name as the input. The resulted representation codes semantic information of each input attribute, so called the attribute semantics.

Class attribute vectors: A set of vectors corresponds to the set of classes. Each vector encodes information about the attributes of the corresponding class, and the presence of attribute is indicated by a continuous or binary value at its corresponding element of the vector.

Attribute prototypes is actually the class attribute vectors in the attribute space, that has identical meanings with class attribute vectors.

