# OpenReview forum: "Attribute Alignment and Enhancement for Generalized Zero-Shot Learning"
_ICLR.cc/2023/Conference — Submitted to ICLR 2023_

### Official Review · Reviewer_hgQc · 2022-10-23

**Confidence:** 4
**Clarity, Quality, Novelty And Reproducibility:** 1. The novelty of this paper is limit…
**Correctness:** 3
**Technical Novelty And Significance:** 2
**Empirical Novelty And Significance:** 2
**Recommendation:** 3

**Strength And Weaknesses:**

Strength:
1. The paper is well-organized, which is easy to read. However, some details are unclear.
2. The performance seems good.
3. The combination of attribute alignment and enhancement seems novel, but the novelty is limited.

Weakness:
1. The novelty of this paper is limited. It seems to be a combination of AL [Yang et al. (2021)] and attribute relation graph[Hu et al. (2022)]. As can be seen in the approach, most parts introduce the workflow without detailed formulation.
2. Eq 2 aims to learn projection from the visual space to the word vector space. However, there is no supervision loss. How to guarantee the projections are properly learned?
3. How does attribute alignment conducted? The supervision seems to be classification losses and no explicit alignment is conducted.
4. What is the difference between Eq 8 and Eq 6? They use the same representations.
5. How does the ‘triangle’ in Eq 10 obtained? It is strange to know whether a sample is seen or unseen.
6. The results in Table 1 are unclear. Why the input size on AWA is different from the other two datasets? Do the authors use different networks?
7. The authors only describe the phenomenon in Section 4.4 without detailed analysis. Moreover, no detailed qualitative analysis to show the effectiveness of the proposed approach.


**Summary Of The Paper:**

This paper proposes an attribute alignment and enhancement (A3E) network for zero-shot learning. It uses the attribute location model to align attributes and utilized the attribute relation graph to enhance the attribute. Experiments on three datasets show the effectiveness of the proposed approach.

**Summary Of The Review:**

The novelty of this paper is limited and the detailed implementations are unclear. The experiments are not sufficient, which lacks experiment analysis and qualitative results to show the effectiveness.

---

> ### Author Response · Authors · 2022-11-19
> **Response to the reviewer hgQc (1)**
>
> Dear Reviewer,
>
>  Thank you very much for your invaluable comments and fruitful suggestions for our paper. We revised the paper carefully based on your comments.
>
> The following is the responses to your comments.
>
> **1. The novelty of this paper is limited. It seems to be a combination of AL [Yang et al. (2021)] and attribute relation graph [Hu et al. (2022)]. As can be seen in the approach, most parts introduce the workflow without detailed formulation.**
>
> Thank you very much for your valuable comments. We feel very sorry for the inadequate presentation of motives in this paper.
>
> Based on our study and reflection on related fields, we believe that relation learning has the potential to address the challenges of GZSL. Although Hu et al. had modeled semantic relationships of attributes based on attribute relation graph [1], it was a static attribute relation modelling method. Here, dynamic modelling of relations is crucial to attribute relation learning. Accordingly, we proposed our A3E network.
>
> The key to A3E network is attribute enhancement based on semantic relationships of attribute, for which we need an efficient and concise way to align attributes with their corresponding visual features. Therefore, we propose an attribute alignment pipeline. Despite its simple structure, AA pipeline is able to effectively fulfil two of the model's requirements, which are locating the visual features of the attributes by AL module and mapping them to the semantic space. More importantly, AA pipeline achieves these tasks in a joint pipeline, making an important contribution to the achievement of the desired model performance. As far as we know, it is the first attempt in this field.
>
> Based on your comments. We have extensively revised the 2nd and 3rd paragraphs for a better presentation of the motivation of the proposed method. The revised part is not included due to characters limit.
>
> Reference:
>
> [1] Hu, Yang, et al. "Graph-based visual-semantic entanglement network for zero-shot image recognition." IEEE Transactions on Multimedia 24 (2021): 2473-2487.
>
> **2. Eq 2 aims to learn projection from the visual space to the word vector space. However, there is no supervision loss. How to guarantee the projections are properly learned?**
>
> Thank you very much for your comments. We apologize for any confusion that may have been caused by the text layout.
>
> Eq (3):${{\bf{\hat e}}_a} = {\bf{W}}\left( {GAP\left( {{{\bf{v}}^{\left( a \right)}}} \right)} \right),a \in \left[ {1,A} \right]$formulates the projection of AVF to word vectors. In order to learn this mapping, we obtain the class score ${s^c}\left( {c \in \left[ {1,{N^S}} \right]} \right)$ via Eq (4) and (5):
>
> ${p_a} = {{\bf{e}}_a}{\bf{\hat e}}_a^T,a \in \left[ {1,A} \right]$
>
> ${s^c} = {{\bf{a}}^c}{{\bf{p}}^T},c \in \left[ {1,{N^S}} \right]$
>
> And concatenate them into class score vector ${{\bf{s}}_{AS}} \in \mathbb{R} {^{{N^S}}}$ to calculate loss (6):
>
> $L_{AS} = {\cal C}{\cal E}\left( {SoftMax\left( s_{AS} \right),{y_i}} \right)$
>
> In fact, the only trainable parameter in above equations is the mapping $\bf{W}$, which is under the supervision of $L_{AS}$.
>
> **3. How does attribute alignment conducted? The supervision seems to be classification losses and no explicit alignment is conducted.**
>
> Thank you very much for your valuable comments. We feel sorry for any misunderstandings that may have been caused.
>
> The proposed Attribute Alignment is a joint pipeline that consist of Attribute Localization module and Attribute Scoring module, where the AL module is responsible for localizing the visual features of the attributes, i.e. spatial alignment. The principle of spatial alignment has been interpreted by Yang et al. in [1]. The Class Activation Map (CAM) [2] shows which part of the image contributes more to the current class prediction, and our attribute mask (the outcome of attribute localization) is a linear combination of CAMs that are actually under the supervision of classification loss. And the AS module further maps the localized visual features (AVFs) to the attribute semantic space, i.e. semantic alignment. Under the supervision of their respective classification losses, we integrated them into a joint pipeline to collaborate on the task, so called the attribute alignment.
>
> Reference:
>
> [1] Yang, Shiqi, et al. "On implicit attribute localization for generalized zero-shot learning." IEEE Signal Processing Letters 28 (2021): 872-876.
>
> [2] Zhou, Bolei et al. “Learning Deep Features for Discriminative Localization.” 2016 IEEE Conference on Computer Vision and Pattern Recognition (CVPR) (2016): 2921-2929.

---

> > ### Author Response · Authors · 2022-11-19
> > **Response to the reviewer hgQc (2)**
> >
> > **4. What is the difference between Eq 8 and Eq 6? They use the same representations.**
> >
> > Thank you very much for your valuable comments. We are sorry for the confusion caused.
> >
> > As we mentioned in Section 3.3:
> >
> > “As the output of AE module has the same form of predicted word vectors like AS module, an integrated enhanced attribute scoring (EAS) module is naturally formed,”
> >
> > In fact, EAS can be considered as an upgraded version of the AS module and both have the same form of output, therefore the same loss function expressions. The difference lies in the content of the resulting predicted word vectors, i.e. with or without the enhancement based on semantic relations.
> >
> > Based on your suggestions, we have revised these equations to avoid further confusion:
> >
> > Eq (8):
> >
> > $L_{EAS} = {\cal C}{\cal E}\left( {SoftMax\left( {s_{EAS}} \right),{y_i}} \right)$
> >
> > Eq (6):
> >
> > $L_{AS} = {\cal C}{\cal E}\left( {SoftMax\left( s_{AS} \right),{y_i}} \right)$
> >
> > where ${{\bf{s}}_{AS}} \in \mathbb{R} {^{{N^S}}}$ is the concatenation of class score ${s^c}\left( {c \in \left[ {1,{N^S}} \right]} \right)$.
> >
> > **5. How does the ‘triangle’ in Eq 10 obtained? It is strange to know whether a sample is seen or unseen.**
> >
> > Thank you very much for your valuable comments. Sorry for the incorrect description and the misunderstanding caused.
> >
> > The statement “…an indicator which will take 1 if the sample is from unseen classes, otherwise takes -1.” is an inappropriate quotation to the articles we referred to [1]. As your comments, we also find it misleading and confusion. In fact, the actual effect of this indicator is to assign 1 to all components of the unseen classes and -1 to all components of the seen classes of the prediction probability vector of a given sample.
> >
> > In accordance to the actual situation in the model, the statement has been revised as:
> >
> > “…an indicator which will take 1 for unseen classes, and -1 for seen classes”
> >
> > References:
> > [1] Huynh, Dat, and Ehsan Elhamifar. "Fine-grained generalized zero-shot learning via dense attribute-based attention." Proceedings of the IEEE/CVF conference on computer vision and pattern recognition. 2020.
> >
> > **6. The results in Table 1 are unclear. Why the input size on AWA is different from the other two datasets? Do the authors use different networks?**
> >
> > Thank you very much for your valuable comments. We feel very sorry for the confusion caused by the layout of the paper. Due to the limitation of paper length, we have moved the implementation details of the network into Appendix C which describes the reasons for choosing different input sizes:
> >
> > “The input images of model are reshaped as 224×224 pixels for AwA2 datasets and 448×448 pixels for CUB and SUN datasets since the finer details could significantly improve performance on fine-grained datasets.”
> >
> > As we tested, AWA2 dataset does not require the additional input size to achieve competitive results, which will lead to unnecessary model costs. In fact, different input sizes will produce different shapes of visual features. However, this difference will not affect the proper operation of the model, as the resulting features will be global pooled to the same shape before the critical step.
> >
> > **7. The authors only describe the phenomenon in Section 4.4 without detailed analysis. Moreover, no detailed qualitative analysis to show the effectiveness of the proposed approach.**
> >
> > Thank you very much for your valuable comments. We apologize for the incompleteness of this section. According to your comments, we have added the analysis to relevant phenomena in Section 4.4, as follows:
> >
> > “4.4.1 EFFECTS OF WEIGHTING COEFFICIENT λ
> >
> > …*The increasing of λ represents that the model emphasizes more on AL module learning. Results show that despite the simpler structure and fewer parameters of the AL module, its importance in the objective function is no less than that of the EAS module. It is proven by the results that model performance improves as the effort invested in AL module (i.e., value of λ) increases. Specially, attribute localization is more demanding for the fine-grained datasets with more subtle features, such as the CUB dataset*....”
> >
> > “4.4.2 EFFECTS OF THRESHOLD δ
> >
> > …*With the increase in δ, the edges representing attribute relationships in ARG should gradually decrease. Obviously, edges created by PMI do not perfectly correspond with the semantic relationships of attributes. We believe that the graphs generated by lower values of δ have more noisy connections (edges), which leads to the model performance decline. When the structure of ARG is simplified with higher threshold, GAT is more likely to obtain robust information from ARG, thus enhancing the classification performance*….”

---

### Official Review · Reviewer_a3kk · 2022-10-23

**Confidence:** 1
**Correctness:** 3
**Technical Novelty And Significance:** 2
**Empirical Novelty And Significance:** 1
**Recommendation:** 5

**Clarity, Quality, Novelty And Reproducibility:**

The paper is presented with a high-quality standard. The idea is clear and easy to follow. However, it is difficult to give credit to the originality since the essential modules are from other work and no significant modification or theoretical analysis can be found.

**Strength And Weaknesses:**

The paper is well-presented and well-designed.
- Localisation, GAT are not new to ZSL. No clear motivation or novel insights for these designs are provided.
- Ablation study is very rough and superficial.
- Experimental results are not competitive.
- Lack of theoretical contributions.

**Summary Of The Paper:**

This paper addresses Zero-Shot Learning problems by localisation and GAT based attribute enhancement. The method is evaluated on the three datasets and did not achieve state-of-the-art results.

**Summary Of The Review:**

Overall, the paper is lack of theoretical contribution and the performance is not promising. Although the paper is well written and presented, it is still below the standard of ICLR expected.

---

> ### Author Response · Authors · 2022-11-19
> **Response to the reviewer a3kk**
>
> Dear Reviewer,
>
>  Thank you very much for your invaluable comments and fruitful suggestions for our paper. We revised the paper carefully based on your comments.
>
> The following is the responses to your comments.
>
> **1. Localisation, GAT are not new to ZSL. No clear motivation or novel insights for these designs are provided.**
>
> Thank you very much for your valuable comments. We feel very sorry for the inadequate presentation of motives in this paper.
>
> Based on our study and reflection on related fields, we believe that relation learning has the potential to address the challenges of GZSL, and GAT is a trusted tool for relation learning, which can dynamically alter node connections, rather than GCN, has a significant impact on relation modelling. And attribute localization is one component of the whole structure. The key to A3E network is attribute enhancement based on semantic relationships of attribute, for which we need an efficient and concise way to align attributes with their corresponding visual features. Therefore, we propose an attribute alignment pipeline. Despite its simple structure, AA pipeline is able to effectively fulfil two of the model's requirements, which are locating the visual features of the attributes and mapping them to the semantic space. More importantly, AA pipeline achieves these tasks in a joint pipeline, making an important contribution to the achievement of the desired model performance. As far as we know, it is the first attempt in this field.
>
> Based on your comments. We have extensively revised the 2nd and 3rd paragraphs of Introduction for a better presentation of the motivation of the proposed method.
>
> **2. Ablation study is very rough and superficial.**
>
> Thank you very much for your valuable comments. We are sorry for the oversimplified ablation study due to length limitations. Our intention was to demonstrate the contribution of the proposed novel attribute enhancement to the model. As your comment, we are aware that there are more advantages of the method that can be demonstrated through ablation studies.
>
> We consider a comparison of the performance between GCN and GAT as the backbone networks of AE module to explore the impact of relational learning for ZSL. We have also designed the experiments using a two-layer linear transformation which is closer to the GAT structure instead of the original single-layer linear transformation (from the DAZLE [1] model) in the baseline model. Unfortunately, since the above experiments are still under operation, we were unable to provide rigorous results on time. We are very sorry that we can only try to interpret the implication of the original ablation experiment.
>
> References:
> [1] Huynh, Dat, and Ehsan Elhamifar. "Fine-grained generalized zero-shot learning via dense attribute-based attention." Proceedings of the IEEE/CVF conference on computer vision and pattern recognition. 2020.
>
> **3. Experimental results are not competitive.**
>
> Thank you very much for your valuable comments. We think seriously about your suggestions and give responses as follows.
>
> We can understand your questions about the results of the experimental comparison of the proposed method. However, comparisons in this field are never easy. The majority of our main comparison models only exceeded their rivals in about half of their benchmark results. Our main competitor, MSDN [1], is the state-of-the-art in this direction that once had the best benchmark scores, while A3E still managed to outperforms it in AWA2 dataset, and shows competitive results in CUB dataset.
>
> Reference:
> [1] Chen, Shiming, et al. "MSDN: Mutually Semantic Distillation Network for Zero-Shot Learning." Proceedings of the IEEE/CVF Conference on Computer Vision and Pattern Recognition. 2022.
>
> **4. Lack of theoretical contributions.**
>
> Thank you very much for your valuable comments. We apologize again for the lack of clear presentation of motives and contributions in the paper.
>
> The core of the proposed A3E model is the innovation of Attribute Enhancement, which leverages attribute semantic relationships to enhance attribute representation.
>
> Based on your comments. We have extensively revised the corresponding paragraphs for a better presentation of the motivation of the proposed method. The following is the revised part. As we stated in the paper, while all samples of each category are assigned to the same class attribute vector, the actual situation regarding attributes will obviously vary from sample to sample. This raises the need for attribute enhancement. And the semantic relations between attributes existing in attribute word vectors provide the feasibility of attribute enhancement. To the best of our knowledge, the proposed method is the first attempt to learn enhanced representation of attributes in attribute semantic space. More than that, the approach of implementing representation enhancement based on relation learning has the potential to be referenced by many more domains.

---

### Official Review · Reviewer_2RdH · 2022-10-25

**Confidence:** 5
**Clarity, Quality, Novelty And Reproducibility:** Limited novelty and unfair comparison…
**Correctness:** 2
**Technical Novelty And Significance:** 1
**Empirical Novelty And Significance:** 2
**Recommendation:** 3

**Strength And Weaknesses:**

The organization is fine. Experiments are sufficient.

However,

1.Many statements in this paper are not well motivated, e.g., ``implicit localization capability within the feature extractor’’ is not clear; ``exciting achievements for common categories’’ is also not clear. The story on comparisons among human and ZSL is commonly used in the community; you cannot say this because others have done this. The resulting facts of ZSL can not guarantee the reason of using auxiliary information for knowledge transferring. In fact, auxiliary information is one choice of conducting ZSL, there much exist other manners for performing ZSL,e.g., matching with external dataset. By saying embedding methods are inherently inferior in GZSL to generative methods, why? Also, it is fine to use unseen semantics for overcoming bias issue, yet not prone to unexpected effects. By saying unexpected effects, what do you mean? Overall, the challenges are not well defined. The writing is also not accurate.

2.Terms of attribute semantics, class attribute vectors, attribute prototypes and attribute-visual features are confusing to the beginner of ZSL field. More explanation should be given. Generative adversarial network (GAN) or variational autoencoder (VAE) miss references. Typos exists: so that convert ?

3.The motivation for the usage of GAT for improving the generalization is not clear. In fact, using graph neural network to model attribute relationships have been explored in previous works (e.g., LsrGAN in ECCV20). The overall compared methods are limited, and more methods should be surveyed and compared.

4.The framework is actually incremental improvements on existing methods, e.g., improved on Yang et al. in CVPR 2021 by seeing each channel as attention mask. From this point, the novelty is limited to the community, since this kind of attention has been explored in the community (e.g., in AREN of CVPR2019). Furthermore, the idea of mapping AVFs into attribute semantic space by DAZLE is also not new. The idea of attribute scoring is just the combination of DAZLE with AREN for solving the same task in this paper. All these aspects reduced the contributions of this paper to the community.

5.The overall writing is poor. The formulas are also confusing. Please amend them accordingly.

6.How to initialize the network weights, by pre-trained ImageNet weights or from scratch? What’s the specific network architecture?

7.Since additional operation such as attention is used for the proposed method, I am doubted about the tradeoff between the running time and accuracy.

8.The parameter analysis is shown, however, what’s the specific parameters for achieving these results? E.g. \beta. It seems the authors have reported adjusted results by varying many \beta, this is thus not fair to compare with other methods without CS adjustments. Also, the authors used additional attribute features for model training which is also not fair compared with counterparts.

**Summary Of The Paper:**

This paper proposes to fully utilize attributes information for GZSL for simultaneously exploring attribute alignment and enhancement (A3E). A3E consists of an attribute localization (AL) module and enhanced attribute scoring (EAS) module. AL is used for localizing attribute regions, and EAS further enhances these features by GAT. Experiments are conducted on golden GZSL datasets.

**Summary Of The Review:**

Combination of existing works, and limited novelty to the community.

---

> ### Author Response · Authors · 2022-11-19
> **Response to the Reviewer 2RdH (1)**
>
> Dear Reviewer,
>
> Thank you very much for your invaluable comments and fruitful suggestions for our paper. We revised the paper carefully based on your comments.
>
> The following is the responses to your comments.
>
> ---------- Responses to the comments----------
>
> **1. （1）Many statements in this paper are not well motivated, e.g., implicit localization capability within the feature extractor’’ is not clear;**
>
> The statement “implicit localization capability within the feature extractor” is a faithful quotation to the articles we referred to [1]. According to [2], CNNs have remarkable ability to localize objects in the convolutional layers. Since the relevant principles have been studied in details, we have decided to retain the statement with clear citation in respect to the original work.
>
> Reference:
>
> [1] Yang, Shiqi, et al. "On implicit attribute localization for generalized zero-shot learning." IEEE Signal Processing Letters 28 (2021): 872-876.
>
> [2] Zhou, Bolei et al. “Learning Deep Features for Discriminative Localization.” 2016 IEEE Conference on Computer Vision and Pattern Recognition (CVPR) (2016): 2921-2929.
>
> **1. (2)exciting achievements for common categories’’ is also not clear. The story on comparisons among human and ZSL is commonly used in the community; you cannot say this because others have done this. The resulting facts of ZSL can not guarantee the reason of using auxiliary information for knowledge transferring. In fact, auxiliary information is one choice of conducting ZSL, there much exist other manners for performing ZSL, e.g., matching with external dataset.**
>
> Thank you very much for your valuable comments. We feel very sorry for the incorrect statements. According to your suggestion, we have revised the incorrect statements and significantly simplified the introduction. The inappropriate descriptions have been trimmed to avoid misunderstanding. The following is the revised part on the first paragraph of Introduction.
>
> “Zero-shot learning aims to recognize unseen classes that have not been appeared during training phase, a common solution resort to auxiliary information to bridge the gap between seen and unseen domains to achieve knowledge transfer from the seen to the unseen. Semantics are the most frequently used auxiliary information for ZSL, either by class descriptions, word vectors (Mikolov et al., 2013) or attributes (Farhadi et al., 2009). A general paradigm (Xie et al., 2019; Zhu et al., 2019; Huynh & Elhamifar, 2020b; Min et al., 2020; Xie et al., 2020; Ge et al., 2021; Liu et al., 2021b; Chen et al., 2021b; 2022) is to learn a mapping that projects visual features of seen samples into an embedding space to align with semantic attributes. With the assumption that seen and unseen domains share the same attribute space, the learned knowledge from seen classes is easily transferred to the unseen ones. And then, the subsequent classification is accomplished by measuring compatibility scores between the projected features and the attribute prototypes. Recent works on embeddings turn to local features of image parts, i.e. part-based embedding methods (Elhoseiny et al., 2017), to learn discriminative features easy for classification. Comparatively, generative methods (Xian et al., 2019b; Huynh & Elhamifar, 2020a; Ma & Hu, 2020; Han et al., 2021; Chen et al., 2021a;c; Chou et al., 2021) utilize semantic information of unseen classes to synthesize unseen visual features by a generative model, such as generative adversarial network (GAN) (Goodfellow et al., 2020) or variational autoencoder (VAE) (Kingma & Welling, 2013), so that convert zero-shot classification to the traditional supervised model learning that could be trainable with generated samples. However, the features inferred from semantic information mostly are high-level visual representation, which are often non-discriminative to class recognition (Huynh & Elhamifar, 2020b; Xian et al., 2019b; Huynh & Elhamifar, 2020a).”

---

> ### Author Response · Authors · 2022-11-19
> **Response to the Reviewer 2RdH (2)**
>
> **1. (3)By saying embedding methods are inherently inferior in GZSL to generative methods, why? Also, it is fine to use unseen semantics for overcoming bias issue, yet not prone to unexpected effects. By saying unexpected effects, what do you mean? Overall, the challenges are not well defined. The writing is also not accurate.**
>
> Thank you very much for your valuable comments. Sorry for the incorrect description and the misunderstanding caused. We wanted to convey the severe performance degradation of embedding methods in the transition from ZSL to GZSL. As you pointed out, the statement has been revised as:
>
> “Embedding methods are inherently inferior in GZSL since the model training merely relies on samples of seen classes, and thus inevitably biases towards the seen ones.”
>
> As your comments, using unseen semantics is a common method for overcoming bias issue. Our original intention was to discuss about the consequences of applying direct calibration to the prediction probabilities based on that method. To avoid further misunderstanding, we have removed the relevant statement from this paragraph in accordance with your suggestion, and left it to the subsequent chapters for further exploration.
>
> **2. Terms of attribute semantics, class attribute vectors, attribute prototypes and attribute-visual features are confusing to the beginner of ZSL field. More explanation should be given. Generative adversarial network (GAN) or variational autoencoder (VAE) miss references. Typos exists: so that convert?**
>
> Thank you very much for your valuable comments. We carefully think about your suggestions and conclude the reasons as follows:
>
> Considering the length limit and the coherence of the text, we tried to keep the descriptions in the paper as restrained and concise as possible. We apologize for the confusion that may have caused. According to your comments, we have included these following explanations of the corresponding terms in the Appendix F and marked them in the paper.
>
> Thank you for the correction, we have included the missing references to the paper [1][2], and the typo has been revised.
>
> Reference:
>
> [1] Goodfellow, Ian, et al. "Generative adversarial networks." Communications of the ACM 63.11 (2020): 139-144.
>
> [2] Kingma, Diederik P., and Max Welling. "Auto-encoding variational bayes." arXiv preprint arXiv:1312.6114 (2013).
>
> Appendix F TERM EXPLANATION:
>
> Attribute semantics: The word vectors of attributes, which is the output of word2vec model by taking the attribute name as the input. The resulted representation is the encoded semantic information of each input attribute, so called the attribute semantics.
>
> Class attribute vectors: A set of vectors corresponds to the set of classes. Each vector encodes information about the attributes of the corresponding class, and the presence of attribute is indicated by a continuous or binary value at its corresponding element of the vector.
>
> Attribute prototypes is actually the class attribute vectors in the attribute space, that has identical meanings with class attribute vectors.

---

> ### Author Response · Authors · 2022-11-19
> **Response to the Reviewer 2RdH (3)**
>
> **3. The motivation for the usage of GAT for improving the generalization is not clear. In fact, using graph neural network to model attribute relationships have been explored in previous works (e.g., LsrGAN in ECCV20). The overall compared methods are limited, and more methods should be surveyed and compared.**
>
> Thank you very much for your valuable comments. We feel very sorry for the inadequate description in the introduction. Based on our study and reflection on related fields, we believe that relation learning has the potential to address the challenges of GZSL, and GAT is a trusted tool for relation learning. To the best of our knowledge, although LsrGAN has explored semantic relationships between classes in depth, it did not involve the study of attribute relations, nor did it employ graph neural networks. However, GVSE [1] is the closest work in this direction and is one of our references, which utilizes GCN to model semantic relationships of attributes. We argue that the use of GAT, which can dynamically alter node connections, rather than GCN, has a significant impact on relation modelling. Experimental results demonstrate that the GZSL performance of proposed method is ahead of GVSE.
>
> We have extensively revised the corresponding paragraphs for a better presentation of the motivation of the proposed method. The revised part is not included in this response deo to the character limits.
>
> For a better demonstration of the competitiveness of our models, we wanted to compare with the best performing models in this field. We have therefore chosen the recent state-of-the-art methods [2][3][4][5] as our main comparisons. Besides, to investigate the effectiveness of the proposed method, we also compared several classical models [6][7] in the same direction.
>
> Reference:
>
> [1] Hu, Yang, et al. "Graph-based visual-semantic entanglement network for zero-shot image recognition." IEEE Transactions on Multimedia 24 (2021): 2473-2487.
>
> [2] Xie, Guo-Sen, et al. "Region graph embedding network for zero-shot learning." European conference on computer vision. Springer, Cham, 2020.
>
> [3] Ge, Jiannan, et al. "Semantic-guided reinforced region embedding for generalized zero-shot learning." Proceedings of the AAAI Conference on Artificial Intelligence. Vol. 35. No. 2. 2021.
>
> [4] Huynh, Dat, and Ehsan Elhamifar. "Compositional zero-shot learning via fine-grained dense feature composition." Advances in Neural Information Processing Systems 33 (2020): 19849-19860.
>
> [5] Chen, Shiming, et al. "MSDN: Mutually Semantic Distillation Network for Zero-Shot Learning." Proceedings of the IEEE/CVF Conference on Computer Vision and Pattern Recognition. 2022.
>
> [6] Zhu, Yizhe, et al. "Semantic-guided multi-attention localization for zero-shot learning." Advances in Neural Information Processing Systems 32 (2019).
>
> [7] Huynh, Dat, and Ehsan Elhamifar. "Fine-grained generalized zero-shot learning via dense attribute-based attention." Proceedings of the IEEE/CVF conference on computer vision and pattern recognition. 2020.
>
> **4. The framework is actually incremental improvements on existing methods, e.g., improved on Yang et al. in CVPR 2021 by seeing each channel as attention mask. From this point, the novelty is limited to the community, since this kind of attention has been explored in the community (e.g., in AREN of CVPR2019). Furthermore, the idea of mapping AVFs into attribute semantic space by DAZLE is also not new. The idea of attribute scoring is just the combination of DAZLE with AREN for solving the same task in this paper. All these aspects reduced the contributions of this paper to the community**
>
> Thank you very much for your valuable comments. According to your comments, As mentioned above, we have extensively revised our introductory section to better present the ideas of the proposed method.
>
> We have checked this point again compared with GEM-ZSL (Yang et al. in CVPR 2021) and have seen the similarity between our approach. However, compared to GEM-ZSL, the proposed attribute alignment has definitely aligned each mask to a certain attribute, which is innovative and crucial to the subsequent attribute enhancement process of this model. In fact, both the attention masks and the mapping of AVFs are dedicated to attribute enhancement. The key to A3E network is attribute enhancement based on semantic relationships of attribute, for which we need an efficient and concise way to align attributes with their corresponding visual features. Therefore, we propose an attribute alignment pipeline. Despite its simple structure, AA pipeline is able to effectively fulfil two of the model's requirements, which are locating the visual features of the attributes and mapping them to the semantic space. More importantly, AA pipeline achieves these tasks in a joint pipeline, making an important contribution to the achievement of the desired model performance. As far as we know, it is the first attempt in this field.

---

> ### Author Response · Authors · 2022-11-19
> **Response to the Reviewer 2RdH (4)**
>
> **5. The overall writing is poor. The formulas are also confusing. Please amend them accordingly.**
>
> Thank you very much for your valuable comments. We are sorry for any of the disruptions caused. We realized that since the EAS and AS modules have the same output form, formula 8 has the exact same expression as formula 6, even though they have different inputs. We have revised these equations to avoid further confusion, and now they can be distinguished by the subscripts of the input prediction vectors:
>
> Eq (8):
>
> $L_{EAS} = {\cal C}{\cal E}\left( {SoftMax\left( {s_{EAS}} \right),{y_i}} \right)$
>
> Eq (6):
>
> $L_{AS} = {\cal C}{\cal E}\left( {SoftMax\left( s_{AS} \right),{y_i}} \right)$
>
> where ${{\bf{s}}_{AS}} \in \mathbb{R} {^{{N^S}}}$ is the concatenation of class score ${s^c}\left( {c \in \left[ {1,{N^S}} \right]} \right)$.
>
> **6. How to initialize the network weights, by pre-trained ImageNet weights or from scratch? What’s the specific network architecture?**
>
> Thank you very much for your valuable comments. We apologize for the lack of details of the model in the text due to space limitations. These details are in fact fully described in the Appendix C MODEL SETTINGS:
>
> “We use the fixed ResNet101 (He et al., 2016) pretrained on ImageNet as the feature extractor of A3E network, which is commonly used as the backbone network in many models (Huynh & Elhamifar, 2020b; Xie et al., 2020; Ge et al., 2021; Chen et al., 2021b; 2022). The input images of model are reshaped as 224×224 pixels for AwA2 datasets and 448×448 pixels for CUB and SUN datasets since the finer details could significantly improve performance on fine-grained datasets. We use the Word2Vec model trained on Google News to generate attribute word vectors with 300 dimensions. We adopt ADAM optimizer (Kingma & Ba, 2017) in model training and set weigh decay to 1×10−5. We empirically set the hidden layers and attention heads of GAT network in EAS module to {200, 1} for AwA2, {200, 4} for CUB and {1000, 5} for SUN, with dropout rate fixed to 0.2.”
>
> **7. Since additional operation such as attention is used for the proposed method, I am doubted about the tradeoff between the running time and accuracy.**
>
> Thank you very much for your valuable comments. We would like to provide you with sufficient information about this, however, most of the comparison models do not involve experiments on time consumption and do not provide reproducible code. Therefore, we can only provide you with the average training times of the proposed method for your reference. All results were based on the model settings recorded in Appendix C
>
> Table Average training times per iteration(epoch)
> | Dataset | AWA2  | CUB   | SUN   |
> |---------|-------|-------|-------|
> | Time(s) | 158.4 | 363.1 | 290.3 |
>
> **8. The parameter analysis is shown, however, what’s the specific parameters for achieving these results? E.g. \beta. It seems the authors have reported adjusted results by varying many \beta, this is thus not fair to compare with other methods without CS adjustments. Also, the authors used additional attribute features for model training which is also not fair compared with counterparts.**
>
> Thank you very much for your valuable comments. We carefully think about your suggestions and conclude the reasons as follows:
>
> All parameters used in the model are actually listed in Appendix C MODEL SETTINGS.
>
> As your comments, we have applied the calibration bias to the method. In fact, most of our mainly compared models were using direct or indirect adjustment on prediction. For example, DAZLE [1], MSDN [2] and RGEN [3], which are among the most successful of the comparisons. While A3E managed to achieve competitive results against them.
>
> From what we know, a significant proportion of our comparison models are making use of attribute semantics (attribute word vectors), such as DAZLE [1], MSDN [2], f-VAEGAN-D2 [3] and Composer [4]. It is natural to leverage attribute semantics and compare with these models.
>
> Reference:
>
> [1] Huynh, Dat, and Ehsan Elhamifar. "Fine-grained generalized zero-shot learning via dense attribute-based attention." Proceedings of the IEEE/CVF conference on computer vision and pattern recognition. 2020.
>
> [2] Chen, Shiming, et al. "MSDN: Mutually Semantic Distillation Network for Zero-Shot Learning." Proceedings of the IEEE/CVF Conference on Computer Vision and Pattern Recognition. 2022.
>
> [3] Xian, Yongqin, et al. "f-vaegan-d2: A feature generating framework for any-shot learning." Proceedings of the IEEE/CVF Conference on Computer Vision and Pattern Recognition. 2019.
>
> [4] Huynh, Dat, and Ehsan Elhamifar. "Compositional zero-shot learning via fine-grained dense feature composition." Advances in Neural Information Processing Systems 33 (2020): 19849-19860.

---

### Decision · Program_Chairs · 2023-01-20

**Decision:**

Reject

**Justification For Why Not Higher Score:**

All three expert reviewers recommend rejection based on incremental novelty (see references provided by the reviewers), unclear motivation, issues with the write up, and performance not competitive with other SOTA methods. The author response was not sufficient to eliminate these concerns.

**Justification For Why Not Lower Score:**

N/A

**Metareview: Summary, Strengths And Weaknesses:**

The paper presents an approach that utilize attributes for the problem of generalized zero-shot learning. While this is an important problem, all three expert reviewers recommend rejection based on incremental novelty, unclear motivation, issues with the writing, and performance not competitive with other SOTA methods.

**Summary Of Ac-Reviewer Meeting:**

N/A